# Mid-latitudinal habitable environment for marine eukaryotes during the waning stage of the Marinoan snowball glaciation

Huyue Song [1] ✉, Zhihui An[2], Qin Ye[1], Eva E. Stüeken[3], Jing Li[1], Jun Hu[1], Thomas J. Algeo[1,4,5], Li Tian[1], Daoliang Chu [1], Haijun Song [1], Shuhai Xiao [6] & Jinnan Tong[1]

During the Marinoan Ice Age (ca. 654–635 Ma), one of the 'Snowball Earth' events in the Cryogenian Period, continental icesheets reached the tropical oceans. Oceanic refugia must have existed for aerobic marine eukaryotes to survive this event, as evidenced by benthic phototrophic macroalgae of the Songluo Biota preserved in black shales interbedded with glacial diamictites of the late Cryogenian Nantuo Formation in South China. However, the environmental conditions that allowed these organisms to thrive are poorly known. Here, we report carbon-nitrogen-iron geochemical data from the fossiliferous black shales and adjacent diamictites of the Nantuo Formation. Iron-speciation data document dysoxic-anoxic conditions in bottom waters, whereas nitrogen isotopes record aerobic nitrogen cycling perhaps in surface waters. These findings indicate that habitable open-ocean conditions were more extensive than previously thought, extending into mid-latitude coastal oceans and providing refugia for eukaryotic organisms during the waning stage of the Marinoan Ice Age.

The two global Snowball Earth events (i.e., the Sturtian and Marinoan ice ages) during the Cryogenian Period (720–635 Ma) played a key role in the evolution of the Earth-life system[1–5]. During Cryogenian Period, the oceanic redox landscape was restructured[6,7], and animals probably emerged[3], highlighting the possible impact of this climatic event on the Earth's biosphere[8,9]. The surface ocean is hypothesized to have been mostly or completely frozen during this glacial event[1,10], but alternative climate models and sedimentary evidence indicate the presence of open marine waters in low-latitude regions during the Snowball glaciations[11–16]. Consistent with these less extreme climate models, recent geochemical data provide evidence of active marine biogeochemical cycling during the glaciation periods[6,17–19], and genomic and paleontological data suggest that eukaryotic life existed

during and survived the Marinoan Ice Age[2,20,21]. Notably, macroscopic eukaryotic fossils of the Songluo Biota are preserved in black shales interbedded with Marinoan glacial sediments in the Nantuo Formation of South China[22] implying the existence of open waters at least intermittently at the end of this ice age, insofar as glacial deposits may have been mostly laid down during the end of the Marinoan glaciation[10]. However, the relationship between the Songluo Biota and contemporaneous oceanic environmental conditions remains largely unknown.

In this work, we employ carbon-nitrogen-iron (C-N-Fe) geochemical proxies to explore the environmental context of the Songluo Biota during the Marinoan Ice Age. Iron speciation, which is sensitive to redox conditions near the sediment-water interface, indicates dysoxic-

[1]State Key Laboratory of Biogeology and Environmental Geology, School of Earth Science, China University of Geosciences, Wuhan 430074, China. [2]Wuhan Center of China Geological Survey, Wuhan 430205, China. [3]School of Earth & Environmental Sciences, University of St. Andrews, St. Andrews KY16 9AL, UK. [4]State Key Laboratory of Geological Processes and Mineral Resources, China University of Geosciences, Wuhan 430074, China. [5]Department of Geosciences, University of Cincinnati, Cincinnati, OH 45221-0013, USA. [6]Department of Geosciences, Virginia Tech, Blacksburg, VA 24061, USA. ✉ e-mail: hysong@cug.edu.cn

anoxic bottom waters in this setting. In contrast, nitrogen isotopic data, which capture conditions in the photic zone where most productivity occurs, demonstrate the operation of an aerobic nitrogen cycle in the surface waters of the Marinoan ocean. Integrated with paleontological data from the Songluo Biota and geochemical data from other Marinoan-age successions, the results of our study show that biogeochemical cycles were active during the waning stage of the Marinoan Ice Age, and that surface waters hospitable to aerobic eukaryotes were widespread, extending into mid-paleolatitude regions.

## Results

### Geological setting and sample description

The study section is located at Songluo (31.6854° N, 110.5989° E) in the eastern Shengnongjia National Forest, Hubei Province, South China (Fig. 1). During the Cryogenian, the Shengnongjia area was located on the northern margin of the Yangtze Block of the South China Craton[23]. The first-order structure in this area is a dome-shaped anticline with the middle Proterozoic Shennongjia Group in the core (Fig. 1b). The Shennongjia Group represents the oldest sedimentary strata exposed in the Shennongjia area. Neoproterozoic strata sit unconformably on the Shennongjia Group and are distributed in a circular belt around the Shennongjia dome. Neoproterozoic strata are unevenly developed in the Shennongjia area, but are more fully represented in the western flank of the Shennongjia dome. These include the Tonian Liantuo Formation; the Cryogenian Gucheng, Datangpo, and Nantuo formations; and the Ediacaran Doushantuo and Dengying formations[24,25]. On the eastern flank of the Shennongjia dome, where the studied Songluo section is located, strata from the Liantuo Formation to the Datangpo Formation are partially or entirely missing, and the Nantuo Formation directly and unconformably sits on the Shennongjia Group or the Liantuo Formation.

Paleomagnetic data indicate that the South China Craton was located at mid-latitudes, perhaps between ~30°N and ~40°N, during the Cryogenian[26]. In South China, the age of the late Cryogenian Nantuo Formation is constrained between 654 Ma[27,28] and 635 Ma[29,30]. At the Songluo section, the ~290-m-thick Nantuo Formation unconformably overlies the Mesoproterozoic Shengnongjia Group, and it conformably underlies the basal Ediacaran cap dolostone of the Doushantuo Formation[22,23]. It consists mainly of massive diamictites with interbedded silty mudstones and black shales (Supplementary Note 1). Diamictites in the Nantuo Formation contains abundant granite, gneiss, and carbonate clasts that can reach decimeters in size (mainly 5–50 cm in maximum dimension)[31], which are distinct from diamictites of the Gucheng Formation, which contains clasts that are generally small in size (mainly 2 mm–5 cm in maximum dimension) and are derived from sandstone, vein quartz, and chert. At the Songluo section, the diamictites underlying and overlying the fossiliferous black shales are similar in clast compositions, and both contain clasts characteristic of the Nantuo Formation[23].

The Songluo Biota, an assemblage of carbonaceous compressions of various benthic macroalgal fossils[22] (Fig. 1c), is mainly preserved in a black shale interval (here termed the Songluo black shale, SBS) in the lower Nantuo Formation, although sparse carbonaceous compressions are also found in another mudstone interval in the upper Nantuo Formation[32]. The fossiliferous black shales at the Songluo section are characterized by low manganese (Mn) and high total organic carbon (TOC) contents. Black shale interbeds have been reported from the Nantuo Formation at several other sections in South China[6,19], although to date no fossils have been reported from those sections.

### Geochemical data of Songluo section

The SBS and adjacent diamictites have differing organic carbon (C)- nitrogen (N)- aluminum (Al) geochemical characteristics (Fig. 2a; Supplementary Table 1). In the diamictites, $\delta^{13}C_{org}$ fluctuates irregularly between −31.0‰ and −28.5‰, $\delta^{15}N_{TN}$ values are < +4.0‰, TOC and total nitrogen (TN) concentrations are mostly below 0.5% and 0.1%, respectively, with one exception of 0.5% TN. Ratios of organic carbon to total nitrogen (C/N) are

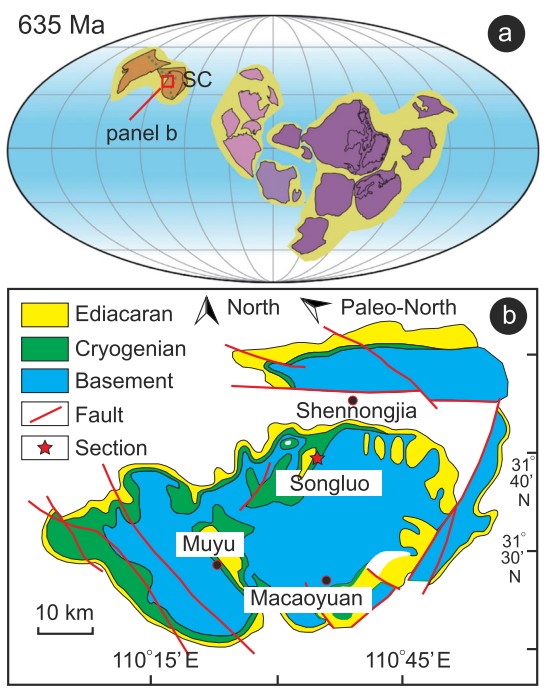

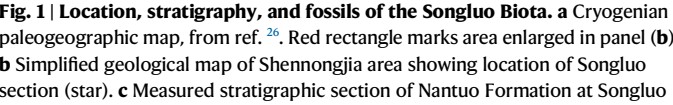

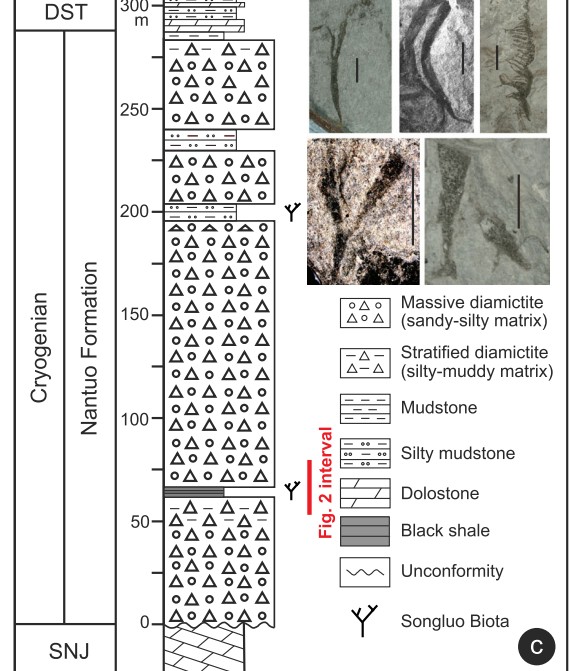

**Fig. 1 | Location, stratigraphy, and fossils of the Songluo Biota. a** Cryogenian paleogeographic map, from ref. [26]. Red rectangle marks area enlarged in panel (**b**). **b** Simplified geological map of Shennongjia area showing location of Songluo section (star). **c** Measured stratigraphic section of Nantuo Formation at Songluo and representative fossils of the Songluo Biota from SBS in lower Nantuo Formation (modified from ref. [32]). SC: South China; SNJ: Shennongjia; DST: Doushantuo; SBS: Songluo black shale. Scale bars: 3 mm.

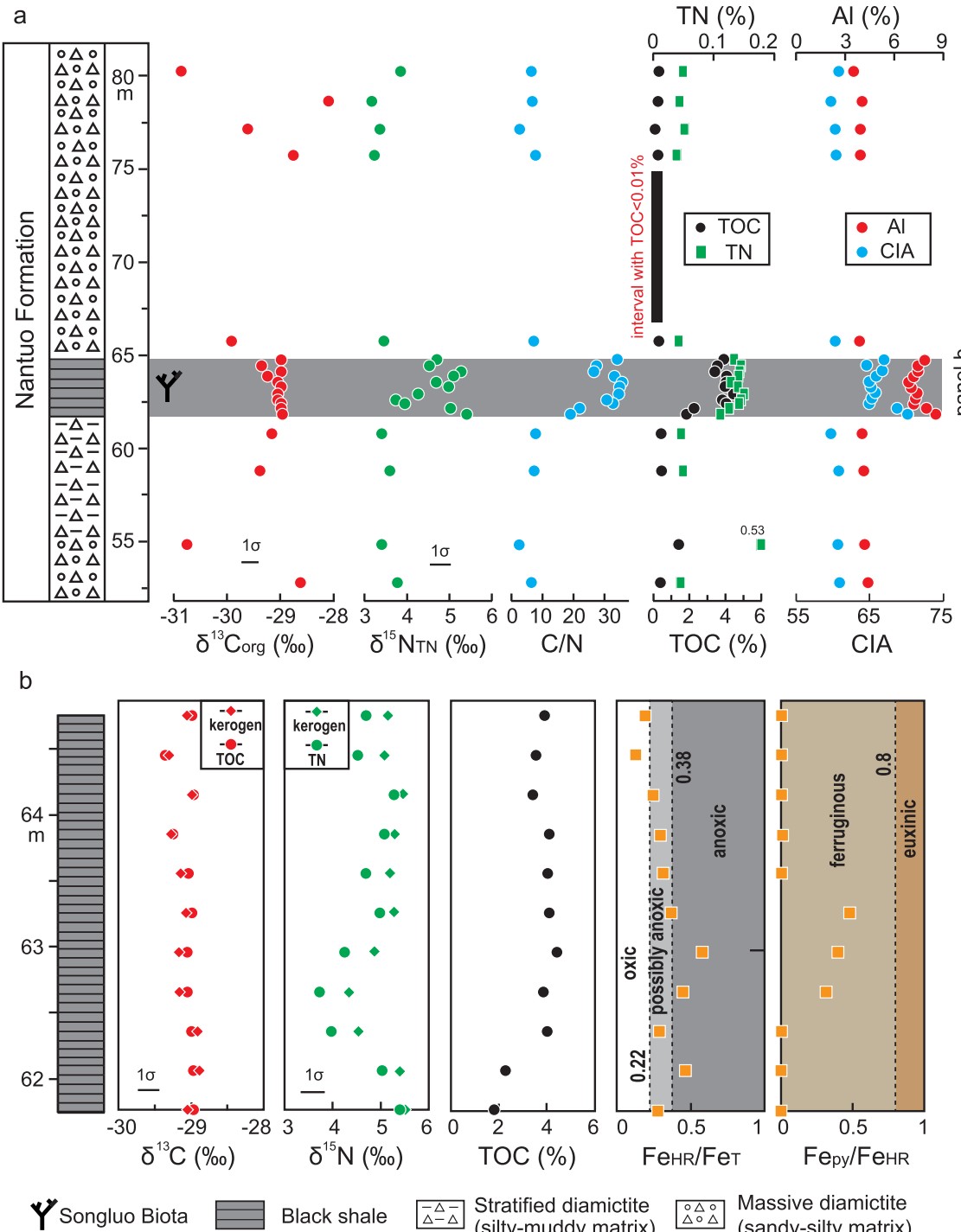

**Fig. 2 | Geochemical data from Nantuo Formation at Songluo. a** $\delta^{13}C_{org}$, $\delta^{15}N_{TN}$, C/N ratios, TOC, TN, Al concentration, and CIA data of the Songluo section. **b** $\delta^{13}C_{org}$, $\delta^{13}C_{kerogen}$, $\delta^{15}N_{TN}$, $\delta^{15}N_{kerogen}$, TOC, $Fe_{HR}/Fe_T$ ratios, and $Fe_{py}/Fe_{HR}$ ratios of Songluo black shale interval that preserves the Songluo Biota. C: carbon; org:

organic; N: nitrogen; TN: total nitrogen; TOC: total organic carbon; Al: aluminum; CIA: chemical index of alteration; $Fe_{HR}$: highly reactive iron; $Fe_T$: total iron; $Fe_{py}$: pyrite iron.

typically <10 (mol/mol), Al concentrations vary between 3.6% and 4.4%, and chemical index of alteration (CIA) values are between 59.7 and 60.7. In the SBS (Fig. 2a, b), both organic carbon isotope values ($\delta^{13}C_{org}$) and kerogen carbon isotope values ($\delta^{13}C_{kerogen}$) are stable between −28.5‰ and −29.5‰, nitrogen isotope values of the bulk rock ($\delta^{15}N_{TN}$) are between +3.5‰ and +5.5‰, kerogen nitrogen isotope values ($\delta^{15}N_{kerogen}$) are between +4.0‰ and +5.5‰, TOC and TN concentrations are above 2% and 0.2%, respectively, C/N ratios vary between 10 and 30, Al

concentrations vary between 6.9% and 8.5%, and CIA values are between 64.6 and 70.2. Nine of the eleven SBS samples have $Fe_{HR}/Fe_T$ ratios >0.22 ($Fe_{HR}$: highly reactive iron; $Fe_T$: total iron), and all samples have $Fe_{py}/Fe_{HR}$ ratios <0.8 ($Fe_{py}$: pyrite iron; Fig. 2b).

## Discussion
The carbon and nitrogen isotopic data of the SBS are stratigraphically consistent and likely record local environmental conditions with a

limited diagenetic alteration. The TOC content of the SBS is much higher than that of adjacent diamictite samples and thus is more buffered against diagenetic alteration and analytical contamination. $\delta^{13}C_{org}$ data of the SBS fall consistently around −29 ‰, similar to those of lower Ediacaran black shales of the Doushantuo Formation in the Yangtze area to the southeast[33], but rather different from those of the adjacent diamictites, which are likely affected by detrital organic matter to various degrees[19]. There is no significant correlation between $\delta^{13}C_{org}$ and TOC (Supplementary Fig. 1a), again indicating minimal diagenetic alteration or analytical contamination of $\delta^{13}C_{org}$. $\delta^{13}C_{org}$ and $\delta^{13}C_{kerogen}$ are similar and well correlated, indicating that the measurements were not affected by incomplete carbonate removal in the laboratory (Fig. 2b, Supplementary Fig. 1e). Also, $\delta^{15}N_{TN}$ and $\delta^{15}N_{kerogen}$ covary (Fig. 2, Supplementary Fig. 1f), although $\delta^{15}N_{kerogen}$ is consistently higher than $\delta^{15}N_{TN}$ by 0.1‰ to 0.6‰, as expected if $^{15}N$-depleted nitrogen was release from organic matter but then recaptured by clay minerals with minimal subsequent metamorphic alteration[34]. These lines of evidence indicate that the nitrogen isotopic data of the SBS have not been notably altered by diagenesis or metamorphism, which would have increased and flipped the divergence between $\delta^{15}N_{TN}$ and $\delta^{15}N_{kerogen}$[35], nor by detrital contamination that would likely have introduced organic-bound and/or silicate-bound N from different sources with distinct compositions.

We note that the C/N ratios of the SBS are much higher than those of Phanerozoic marine sediments, which typically have C/N ratios of 4–10[36]. In post-Ordovician sedimentary rocks, C/N ratios can be elevated by terrestrial organic matter fluxes[36], including organic matter derived from terrestrial plants, which became ecologically important since the Ordovician[37]. Precambrian sedimentary rocks typically have higher C/N ratios[38], but they are unlikely to have been influenced by terrestrial organic matter fluxes; instead, differential preservation of organic carbon vs organic nitrogen, as well as a biological response to high productivity and hence N limitation[39], may have contributed to the high C/N ratios in Precambrian sedimentary rocks. Despite the possibility of differential preservation of organic carbon vs organic nitrogen, the consistent carbon and nitrogen isotope values in the SBS, as well as the lack of correlation between C/N ratios and $\delta^{13}C$ or $\delta^{15}N$ values (Supplementary Fig. 1c, d), imply that organic carbon decomposition and preservation did not substantially modify $\delta^{13}C$ or $\delta^{15}N$ values. In contrast, phyllosilicate-bound ammonium from detrital material may have contributed to the low C/N ratios in the diamictite relative to the SBS. In fact, the $\delta^{15}N_{TN}$ values of Nantuo diamictites in this study are similar to those of Datangpo Formation[40], indicating a possible detrital origin from the latter. Because of the possible influence of detrital material on Nantuo diamictite, the following discussion will be focused on the SBS.

Sedimentary iron speciation has been widely used to distinguish oxic, ferruginous and euxinic conditions in fine-grained siliciclastic rocks[41,42]. $Fe_{HR}/Fe_{T}$ ratios >0.38 and <0.22 represent anoxic and oxic conditions, respectively, with intermediate values (0.22–0.38) recording possible anoxia. In anoxic facies, $Fe_{py}/Fe_{HR}$ ratios are high (>0.7–0.8) under euxinic conditions, in which $Fe_{HR}$ is readily converted to $Fe_{py}$ through reaction with $H_2S$, and low (<0.7–0.8) under ferruginous conditions[42,43]. In the SBS, the $Fe_{HR}/Fe_{T}$ ratios (mainly >0.22) and $Fe_{py}/Fe_{HR}$ ratios (<0.7) indicate that dysoxic-anoxic and ferruginous conditions prevailed in contemporaneous bottom waters in shallow-marine facies at South China continental shelves.

There are limited Fe speciation data from the black shale samples of the Marinoan glacial interval. To the best of our knowledge, the only published data came from the black shales and diamictites in the Ghaub Formation of the Tsumeb Subgroup of the Congo Craton, which is contemporaneous with the Nantuo Formation and also represents deposition during the Marinoan Ice Age[17]; additional data came from the Mineral Fork Formation in Utah[18], although it is uncertain whether this unit was deposited during the Marinoan or

Sturtian age[44]. Fe speciation data from both the Ghaub and Mineral Fork formations indicate predominantly anoxic, ferruginous depositional conditions throughout the successions[17,18]. Recently, Shen et al[6]. published Fe speciation data from diamictite matrix, sandstone, mudstone and shales in the Nantuo Formation, which indicate fluctuating redox conditions between oxic and anoxic states across the Marinoan Ice Age. These results are consistent with other Fe-speciation studies suggesting that predominately oxic surface waters and ferruginous bottom waters characterized the Neoproterozoic oceans[45].

Nitrogen isotope compositions in sedimentary rocks ($\delta^{15}N_{sed}$) are routinely used for reconstructing nitrogen biogeochemical cycling and the evolution of paleoenvironmental redox conditions[40]. $\delta^{15}N_{sed}$ reflects the isotopic composition of buried biomass, which in turn depends on the balance of N inputs ($N_2$ fixation with minimal isotopic fractionation) and outputs (denitrification and anammox with strong isotopic fractionation and removing light N from the ocean)[46]. After the burial, some ammonium is released from degrading biomass and may be trapped as ammonium in clay minerals, while some nitrogen remains organic-bound. Most of the nitrogen isotope studies employed the $\delta^{15}N$ of the bulk rock ($\delta^{15}N_{TN}$) to reconstruct the ancient nitrogen cycle[17,18,47], and there is limited understanding of the difference of the $\delta^{15}N_{TN}$ and the $\delta^{15}N_{kerogen}$[34,35,40]. Our data from the SBS show that $\delta^{15}N_{kerogen}$ is consistently higher than $\delta^{15}N_{TN}$ by 0.1–0.6 ‰, which is typically seen only in unmetamorphosed sedimentary rocks[35]. The decomposition of the organic matter preferentially releases $^{15}N$-delepted ammonium ions during early diagenesis, leading to the enrichment of $^{15}N$ in the residual organic matter. The isotopically lighter ammonium is taken up by clay minerals. With progressive metamorphism, this relationship reverses, possibly due to difference in bond strengths that lead to equilibrium isotope fractionation between kerogen and phyllosilicates[35,40]. Hence our data are consistent with a very low degree of metamorphic alteration where this reverse effect has not yet occurred.

The N-isotopic systematics of the Nantuo Formation provide evidence of an active aerobic nitrogen cycle in the Marinoan ocean where and when the SBS was deposited. Despite the small but consistent offset between $\delta^{15}N_{TN}$ and $\delta^{15}N_{kerogen}$ of the SBS, both proxies fall in a narrow range (+3.7‰ to +5.5‰) that approaches the $\delta^{15}N$ of the modern oceanic nitrate pool[46]. In modern oceans, ammonium from the $N_2$ fixation process is quantitatively oxidized to nitrate ($NO_3^-$) without significant net isotopic fractionation[40]. Within oxygen minimum zones (OMZs), dissolved nitrate is lost from seawater via microbial denitrification and microbial anammox processes, which release $^{14}N$-enriched $N_2$, and result in a $^{15}N$-enriched residual nitrate pool in the ocean[46]. This value is recorded by nitrate-assimilating biomass and thus archived in sedimentary $\delta^{15}N$. Hence, moderately positive $\delta^{15}N_{TN}$ and $\delta^{15}N_{kerogen}$ of the SBS indicate active nitrogen biogeochemical cycling that involved nitrification and denitrification under dysoxic-anoxic conditions during the time of the Songluo Biota (Fig. 3c). The nitrate reservoir likely persisted in oxic surface waters above oxygen-depleted bottom waters. This active nitrogen biogeochemical cycling is different from the $NH_4^+$-dominated nitrogen cycling under the stagnant ocean anoxic condition during the global snowball Earth conditions (Fig. 3a, b).

In contrast, Nantuo Formation diamictites show $\delta^{15}N_{TN}$ values (~3‰) lower than those in the SBS. Similarly, with rare exceptions, most samples from the Marinoan-aged Ghaub Formation in Namibia have $\delta^{15}N_{TN}$ values of 1–4‰[17]. On the other hand, $\delta^{15}N_{TN}$ values are high (4–8‰) in the Mineral Fork Formation of northern Utah and are still higher (6–9‰) in the overlying Kelley Canyon Formation[18], although the depositional age of these units is poorly constrained and the Mineral Fork Formation may represent either the Sturtian or Marinoan Ice Age[44]. Here, most of the studied samples are from the finely bedded units interbedded with massive diamictite in the Ghaub and Mineral Fork Formation[17,18]. Regardless, at face value (and if detrital

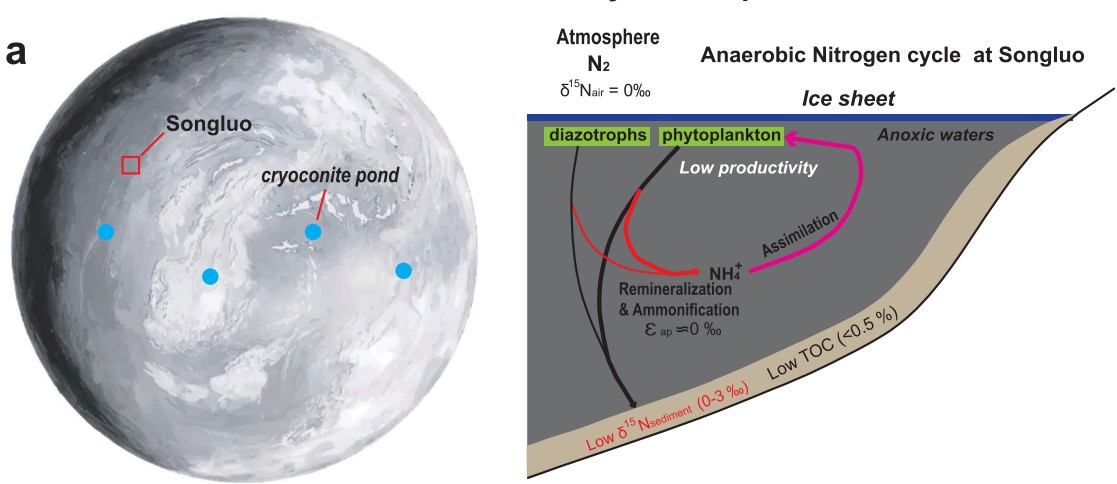

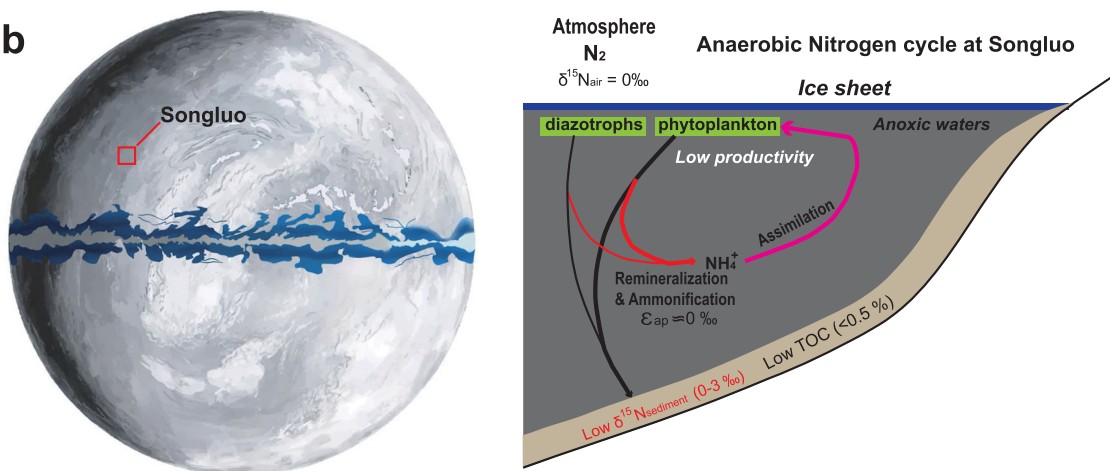

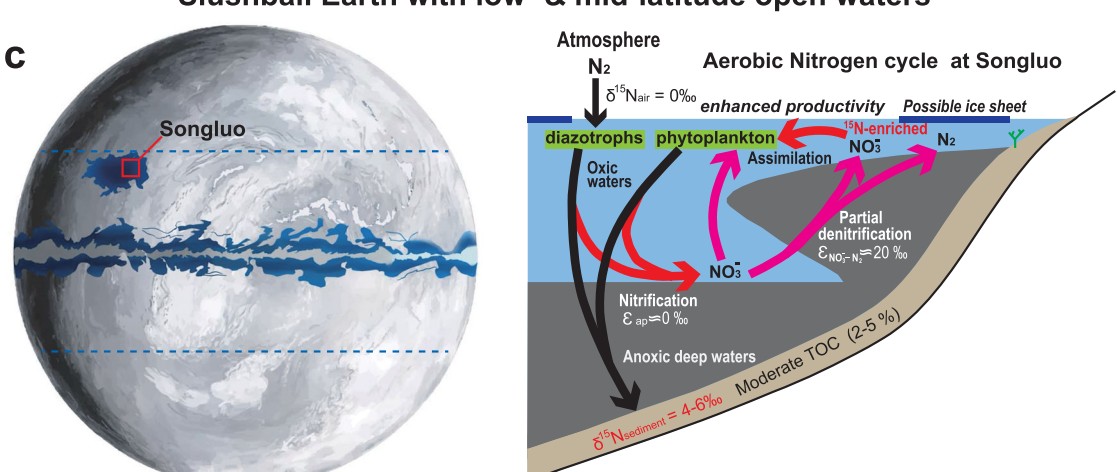

**Fig. 3 | Three models of refugia during Cryogenian Snowball Earth glaciation (left) and simplified model of marine N biogeochemical cycle in mid-latitude oceans during terminal Marinoan Ice Age (right). a** Hard Snowball model, in which refugia are limited to cryoconite ponds[10,49]. **b** Slushball model with a low-latitude open-water belt[12,14,16]. **c** "Slushball model", in which open waters existed in both low- and mid-latitude oceans (this study). Marine N biogeochemical cycle models are modified from ref. [40]. Also note the existence of habitable shallow-water environments for benthic phototrophic macroalgae. Dashed lines in the left panel bracket approximate latitudes where open waters may have existed in shallow oceans. TOC: total organic carbon; N: nitrogen.

contamination can be ruled out), $\delta^{15}N_{TN}$ data from the Nantuo and Ghaub formations indicate that the N-cycle was not purely dominated by $N_2$-fixation, which would generate values around 0‰. Instead, some degree of redox cycling must have taken place that generated nitrate and pushed $\delta^{15}N$ to positive values. This was possibly enhanced at the time of the Songluo Biota. We note that diamictite deposition typically occurred during the deglaciation stage, and the inferred strength of nitrogen recycling applies to the end of the Marinoan Ice Age rather than the entire glaciation.

The co-evolution of life and Earth-surface conditions during the Marinoan Ice Age remains poorly understood. In the "hard Snowball Earth" model, the surface ocean is hypothesized to have frozen over for tens of millions of years, hampering oxygen and nutrient exchange with the atmosphere and delivery from continents, and producing an anoxic ocean that was hostile to aerobic eukaryotes[1] (Fig. 3a). However, several climate models have inferred the presence of an open-water belt in low latitudes[12,14–16], thus providing a refugium for aerobic eukaryotes (Fig. 3b). Others have argued that cryoconite holes or cryoconite pans—where the accumulation of dust and soot reduced local albedo, facilitated heat absorption, and promoted ice melting—supported supraglacial oligotrophic meltwater ecosystems for cyanobacteria and aerobic eukaryotes[10,48,49] (Fig. 3a). Furthermore, organic carbon isotopes ($\delta^{13}C_{org}$) and pyrite sulfur isotopes ($\delta^{34}S_{py}$) from Nantuo Formation diamictite and sandstone/siltstone/mudstone indicate the presence of active carbon and sulfur cycles during the Marinoan Ice Age[19]. Similarly, iron speciation and nitrogen isotope data from the age-equivalent Ghaub Formation provide additional geochemical evidence for active nitrogen cycles during the Marinoan Ice Age[17]. The Fe speciation and nitrogen isotope data presented here from the SBS, which also preserves the Songluo Biota[22], add further and even stronger evidence for a complex, aerobically-based ecosystem in the surface ocean, at least episodically during the end of the Marinoan Ice Age, insofar as the Nantuo Formation was likely deposited during the waning stage of the glaciation.

Our findings have two significant implications. First, the SBS paleontological and geochemical data indicate that ecological refugia for aerobic eukaryotes may have existed in mid-latitude shallow oceans of the late Marinoan Ice Age, given that the Yangtze Block was located at paleo-latitudes of ~30°–40° N (ref. [26]) (Fig. 3c). This contrasts with previous "Slushball Earth" models that argue for an open-water belt within 5–15° (ref. [14]) or 20–30° (ref. [16]) from the equator (Fig. 3b), and suggests a much wider open-water belt instead. Some climatic modeling studies have inferred that a hard Snowball Earth is difficult to produce under realistic model parameterizations, and that the most likely sets of environmental conditions lead to partially glaciated "Slushball Earths"[12,50]. Although we do not have any climatic modeling results for our new "Slushball Earth" model (Fig. 3c), geological evidence supports the possible existence of open-water conditions existed in the mid- and low-latitude oceans during the Nantuo Ice Age. There are several lithological cycles, each represented by massive diamictite, sandstone/siltstone, mudstone, shales, in the Nantuo Formation of South China. These lithological cycles may represent the advance-retreat cycles of glaciers[6,23,51], indicating the dynamic nature of the Marinoan glaciation, at least during its waning stage when the Nantuo Formation is believed to have been deposited. Similar cyclic deposition of Marinoan units has also been found in other regions, such as Central Siberia[52], Oman[13], Norway[53], southwestern China[54], South Australia[55], and Scotland[56]. These studies suggest that the Marinoan glaciation was dynamic, and that there may have existed open-water conditions in the low and middle latitudes (as bracketed by two dashed lines in Fig. 3c). Second, although our iron data indicate that bottom waters were mostly anoxic, the moderately high $\delta^{15}N_{TN}$ values from the SBS imply a redox-active nitrogen cycle and the presence of a nitrate reservoir in the surface ocean that may have facilitated primary productivity, especially among eukaryotic algae that were unable to fix $N_2$ themselves.

Localized refugia such as supraglacial cryoconite holes and ponds are unlikely to have provided sufficient organic matter to sustain global C and N cycles, given the ephemeral nature and limited expanse of these refugia. This argument may also apply to an extended interval of the Marinoan Ice Age beyond the Songluo Biota, as glacial deposits of the Nantuo and Ghaub formations are also characterized by consistently positive $\delta^{15}N_{TN}$ values, even though these values are not as high as those in the SBS. If so, habitable refugia for aerobic eukaryotes in surface waters may have been more widespread and more sustainable than previously thought and may have allowed a rapid rebound of the biosphere after the Marinoan Snowball Earth episode[57].

## Methods

We collected 11 samples from the SBS and 9 samples from the adjacent diamictites. To prepare these samples for geochemical analysis, they were trimmed to remove veins and surficial weathered rinds, and then ground to powders (<74 μm) using a tungsten ball mill.

### Organic carbon isotope and nitrogen isotope analyzes

For the total organic carbon (TOC) and total nitrogen (TN) analyzes, ~10 g of powder was decarbonated using 3 mol/L HCl at room temperature for 24 h. Insoluble residues were then rinsed with 18.2 MΩ/cm deionized water until neutral, dried at 45 °C overnight, and powdered again.

For the kerogen extraction, ~15 g of powder was added to a Soxhlet extractor to remove extractable organics using dichloromethane. The extracted samples were transferred to polytetrafluoroethylene beakers and soaked in distilled water for 4 h. The samples were subsequently treated with 6 mol/L HCl and 40% HF to remove carbonate minerals and most silicate phases. The acid treatment process was divided into five steps: (1) At a ratio of 6 mL of HCl per 1 g of sample, 6 mol/L of HCl was added to the beaker while stirring. The mixture was stirred at 60 °C for 2 h to fully remove the carbonate and then rinsed with distilled water. Completely removing the supernatant to reduce the abundance of cations (e.g., $Ca^{2+}$) released during the acid treatment can decrease the production of fluorides during the subsequent HF treatment. (2) Then a mixture of 6 mol/L HCl and 40% HF was added into the beaker at a ratio of 2.4 mL HCl and 3.6 mL HF per 1 g of sample. The mixture was stirred at 60 °C for 2 h and then the acid was removed. The samples were rinsed with 1 mol/L HCl three times, and the supernatant was removed. (3) At a ratio of 6 mL HCl per 1 g of sample, 6 mol/L HCl was added into the beaker while stirring. The mixture was stirred at 60 °C for 1 h and then the acid was removed. The samples were rinsed with 1 mol/L HCl three times and the supernatant was removed. (4) The second step was repeated but with an increased stirring time of 4 h. (5) Lastly, the third step was repeated. Finally, the samples were rinsed with distilled water until neutral and the supernatant was again removed. Subsequently, the samples were centrifuged at 1500 x $g$ and dried in an oven at 50 °C. The dried samples were cleaned again with dichloromethane in a Soxhlet extractor to remove soluble organic matter. The resulting kerogen residue was air-dried and then thoroughly dried in an oven at 50 °C, followed by weighing and grinding.

The TOC and TN contents of samples were measured on carbonate-free powders using a Vario Macro Cube elemental analyzer (EA, Elementar, Hanau, Germany) in the State Key Laboratory of Biogeology and Environmental Geology (BGEG) at China University of Geosciences (Wuhan). About 200 mg of powder and 50 mg of tungsten trioxide were packed into a 35 × 35-mm tin capsule. The measured carbon and nitrogen contents were used to calculate whole-rock TOC, TN and TS based on the weight ratio of the pre-digestion sample to the acid-insoluble residue. Standard deviations for carbon, nitrogen and sulfur contents are <0.05 wt.% (1σ) based on replicate analyzes of multiple samples. Thus, the analytical precision of TOC and TN measurements was always better than 0.05 wt.% but varied from sample to

sample depending on carbonate content. Analysis of a blank consisting of 50 mg of tungsten trioxide (Macklin, Shanghai, China, PN T818835) yielded no detectable C and N signals.

For organic carbon isotopic analyzes ($\delta^{13}C_{org}$ and $\delta^{13}C_{kerogen}$), dried and ground decarbonated powder (5–20 mg) or ground kerogen powder (~ 1 mg) were weighed into tin capsules and combusted at 960 °C. The evolved $CO_2$ was analyzed using a Flash HT 2000 Plus elemental analyzer (EA) and continuous-flow Delta V Advantage isotope ratio mass spectrometer (IRMS, Thermo Fisher Scientific) in BGEG. International reference standards USGS40 ($\delta^{13}C_{VPDB}$ = –26.39‰) and IVA-Urea ($\delta^{13}C_{VPDB}$ = –37.32‰) were analyzed along with unknowns for a two-point calibration. Analytical precision was better than 0.2‰ (1σ) for $\delta^{13}C$ based on replicate analyzes of these standards.

For nitrogen isotopic analyzes ($\delta^{15}N_{TN}$ and $\delta^{15}N_{kerogen}$), ground carbonate-free powder (40–80 mg) or ground kerogen powder (~20 mg) were mixed with $V_2O_5$, sealed in a tin capsule, combusted at 1020 °C, and the generated $CO_2$ in the outflow gas mixture was absorbed by an alkali lime trap. Nitrogen isotope results are reported using the standard δ notation as deviations from the $\delta^{15}N$ composition of atmospheric $N_2$ ($\delta^{15}N_{AIR}$ = 0‰). Standard reference materials USGS40 ($\delta^{15}N_{AIR}$ = –4.52‰) and IAEA-N-2 ($\delta^{15}N_{AIR}$ = +20.3‰) were used for $\delta^{15}N$ calibration with a precision better than 0.5‰. Analysis of a 10 mg $V_2O_5$ blank (Thermo Scientific, Cambridge, UK, PN 33837510) yielded no detectable N-isotope signal.

### Iron speciation analysis
Iron speciation analysis was undertaken following the protocol of ref. [58]. Sequential chemical extractions were used to separate sedimentary iron into several highly reactive ($Fe_{HR}$) pools: iron oxides ($Fe_{ox}$), magnetite ($Fe_{mag}$), carbonate-associated iron phases ($Fe_{carb}$), and pyrite ($Fe_{py}$). First, sodium acetate was used to extract $Fe_{carb}$, then dithionite to extract $Fe_{ox}$, and lastly ammonium oxalate to extract $Fe_{mag}$. The concentrations of these Fe species were measured with a Perkin-Elmer DRC-e inductively coupled plasma mass spectrometer (ICP-MS) in BGEG. $Fe_{py}$ content was calculated based on pyrite sulfur concentrations from the chromium reduction analysis assuming pyrite stoichiometry[59]. Total iron ($Fe_T$) was determined by ICP-MS after chemical digestion. Analytical precision was better than 5% for Fe concentrations based on replicate analyzes of extractions at all steps. The nonreactive Fe fraction was determined by difference, i.e., $Fe_T$ minus $Fe_{HR}$.

### Major element and CIA analyzes
The concentrations of major elements in whole-rock samples were measured on a Zsx Primus II wavelength dispersive X-ray fluorescence spectrometer (XRF, RIGAKU, Japan) in Wuhan SampleSolution Analytical Technology Co., Ltd. Chinese national standards (GBW0701-14; GBW07401-08; GBW07302-12) were used for calibration. Analytical precision was better than 2% based on replicate analyzes of standards and samples.

Chemical index of alteration (CIA) is calculated from the ratio of Al to major bases:

$$CIA = Al_2O_3/(Al_2O_3 + K_2O + Na_2O + CaO^*) \times 100. \quad 1$$

where all variables represent the molar amounts of major-element oxides. $CaO^*$ represents the fraction of CaO in silicate minerals, which is used in place of total CaO in order to avoid contributions of CaO from carbonate and phosphate minerals that are not tightly linked to weathering processes[60].

### Data availability
The authors declare that all data supporting the findings of this study are available within the paper and its supplementary file.

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

## Acknowledgements

We would like thank Professor Chao Li (now at Chengdu University of Technology) for the support of the iron species analyzes in his lab at BGEG. We thank Yong Du and Teng Xing for lab assistance. This research was supported by the National Natural Science Foundation of China (42172032; 41872033) to H-Y.S., and the China Geological Survey (1212011120787, 12120114066301) to J.T.

## Author contributions

H-Y.S. designed this research; J.T., Z.A., Q.Y., J.H., L.T., H-J.S., H-Y.S., D.C., and S.X. carried out fieldwork and collected the samples; H-Y.S., Z.A. and J.L. performed the geochemical analysis; H-Y.S., S.X., T.J.A., and E.E.S. developed the manuscript with valuable contributions from other co-authors.

## Competing interests

The authors declare no competing interests.
