## [Peer Review File · Nature Communications]

Mid-latitude habitable environment for marine eukaryotes during the waning stage of the Marinoan snowball glaciationReviewer #1 (Remarks to the Author):

In this study, Song et al., analyzed Fe speciation, organic C and N isotopes of black shale within the Nantuo Formation in the Songluo section, South China. This black shale layer yields diverse macroscopic algae fossils, presumably representing a refuge in a Snowball Earth glaciation. N isotopes strongly argue active N and biological cycles in the interglacial interval during the Marinoan Snowball Earth, consistent with a previous study by Ma et al., (GCA, 2022) based on pyrite S and organic C isotopes. The key point in this manuscript is to combine with the paleogeographic reconstruction of South China, arguing the presence of open ocean in mid-latitude. This argument is different from either the hard Snowball Earth or Water belt models. I think the evidence in supporting this conclusion is solid, and the manuscript is overall well written. I would recommend a minor revision.

There are some issues that need to be addressed in the revision.

(1) I agree with authors that there was open ocean in mid-latitude South China Block during the Nantuo glaciation, but why and how such climatic status could exist?

Although it is not easy to resolve, or even beyond the scope of this ms., authors should at least mention or discuss this point in the discussion. For example, discussions of existing climatic models might be useful..

(2) The geological background and the interpretation of fossiliferous black shale are not well stated in the manuscript. Although the stratigraphy and sedimentological analyses of the Nantuo Formation have been discussed in previous studies, it is still necessary to display a detailed discussion about the correlation of this black shale layer. I think most readers would be more interested in this story than sedimentological papers. Thus, please do not assume all readers know Nantuo Formation.

(3) It looks like a negative correlation between D14N and Fepy/FeHR (Fig. 2b), and there is no pyrite in the lower and upper part of the section. Could pyrite be oxidized?

(4) Only a 'blue hole' in mid-latitude looks odd (Fig. 3). I understand that this is the only report showing the presence of open ocean in mid-latitude. I hope authors could make some revision, although I do not have a better solution (sorry!). In fact, I think Figure 4 is not that informative.

Reviewer #2 (Remarks to the Author):

Please see attached.

Reviewer #2 Attachment on the following page

Review of Habitable oceanic environment for early eukaryotes extended to mid-latitude at the end of the terminal Cryogenian Snowball Earth

Summary

This study provides nitrogen and carbon concentrations/stable isotope values, in addition to iron speciation, to suggest the presence of low-oxygen bottom water with oxygenated upper water. The studied units also contain preserved fossils, the Songluo Biota, which are reported to be macroscopic eukaryotes. The authors suggest that the presence of

Major points

The discussion of diamictite data should note more clearly that the C and N data most likely reflects a signal of erosion. In addition, while there was some discussion of detrital influence in the bedded unit, there were no geochemical tests of this detrital influence. Measuring Al, for example, can help with this.

There are no geochemical measurements to corroborate the interpretation of oxygenated surface water. Enriched N isotope values are certainly consistent with this, but on their own do not robustly demonstrate them.

The dataset is a nice addition to the growing global dataset of geochemistry from the Marinoan glaciation. Yet, the discussion is a bit too wide-ranging and expansive. How did the authors decide where to put the “open water” section in Fig 4c? Could the water belt just be wider at this time? The interpretations are reasonable, but without including previously published data and location/age information it is difficult to say how global the interpretation of oxygenated water is.

Methods.

Does WO3 have any trace C or N in it? Did the authors run samples of just this reagent to test for possible contamination? Same comment for V2O5, as previous work (Brauer and Hahne, 2005) have demonstrated that some V2O5 has trace N.

Minor points

Line 51-54: Why does the presence of eukaryotic fossils imply open water? Are these interpreted as photosynthetic?

Lines 79-83: What is the source of the diamictite? Do the C and N values just reflect provenance?

Line 92: High TOC is not, on its own, a reason to suggest it is resilient to diagenesis. The other metrics, such as lack of correlation between $\delta^{13}\text{C}$ and TOC are much stronger.

Line 146: How much can we rely on Fe-speciation in a diamictite? This should probably just reflect the erosional signal from whatever the provenance is. I don't think you need the end of this paragraph, as fin

Line 176: Add “net” before “isotopic fractionation”, as there is an isotope effect during nitrification.

Line 185: In this paragraph, comparing the current work to previous work, it is important to note the units in the Ghaub and Mineral Fork formation are not diamictite, but rather finely bedded units interbedded within a massive diamictite.

Methods.

Does WO₃ have any trace C or N in it? Did the authors run samples of just this reagent to test for possible contamination? Same comment for V₂O₅, as previous work (Brauer and Hahne, 2005) have demonstrated that some V₂O₅ has trace N.

Figure 2. What are the uncertainties on Fe speciation data?

Figure 3. Do you think it's really like an oxygen minimum zone? Would a more continuous redox-cline also be a possibility?

Figure 4. Can you somehow combine figure 3 and 4? Could the right-hand columns basically replicate what's already in figure 3? This could be a nice summary, instead of two separate figures.

Response to the referees (NCOMMS-22-40658)

Reviewer #1 (Remarks to the Author):

In this study, Song et al., analyzed Fe speciation, organic C and N isotopes of black shale within the Nantuo Formation in the Songluo section, South China. This black shale layer yields diverse macroscopic algae fossils, presumably representing a refuge in a Snowball Earth glaciation. N isotopes strongly argue active N and biological cycles in the interglacial interval during the Marinoan Snowball Earth, consistent with a previous study by Ma et al., (GCA, 2022) based on pyrite S and organic C isotopes. The key point in this manuscript is to combine with the paleogeographic reconstruction of South China, arguing the presence of open ocean in mid-latitude. This argument is different from either the hard Snowball Earth or Water belt models. I think the evidence in supporting this conclusion is solid, and the manuscript is overall well written. I would recommend a minor revision.

Response: Thanks for these positive comments.

There are some issues that need to be addressed in the revision.

(1) I agree with authors that there was open ocean in mid-latitude South China Block during the Nantuo glaciation, but why and how such climatic status could exist? Although it is not easy to resolve, or even beyond the scope of this ms., authors should at least mention or discuss this point in the discussion. For example, discussions of existing climatic models might be useful..

Response: Thanks for this comment. We have added citations to climate models that generated low- to mid-latitude open waters, such as Hyde et al. (2000) and Micheels and Montenari (2008). There is no exact answer as to what factors would have allowed such open waters to exist—all simulations are strongly dependent on the boundary conditions and parameterization chosen for a given model. Hyde et al. (2000) pointed out the general difficulty of creating globally ice-bound conditions using realistic parameterizations, noting that continents tend to glaciate before oceans, and that complete continental glaciation would require an ice volume nearly 10 times the Pleistocene maximum. Micheels and Montenari (2008) showed that it is difficult to achieve a hard Snowball condition unless average global temperatures are -70 °C or lower. Both papers noted strong reductions in atmospheric vapor transport as temperatures fall, making it difficult to sustain massive glaciation. The general conclusion of these studies is that a hard Snowball Earth is difficult to produce, and that most likely sets of environmental conditions lead to partially glaciated “Slushball Earths”. All previous climatic modeling studies were based on the published hypothesis of the hard snowball model or the water-belt snowball earth models. The stagnant hard snowball model (with ponds) hypothesized that the global surface ocean was mostly or completely frozen to a significant depth through the Marinoan glacial event. However, low-latitude open-water belt snowball model indicates the presence of open ocean at low latitude area during the Snowball Earth, implying the glaciations were dynamic, which argued against stagnant glaciations in the hard snowball model. Now, although we do not have any climatic modeling results for our new low- and mid-latitude open water snowball model, the geological evidence supports there

might exist open-ocean condition at the mid- and low-latitude areas during the Nantuo glaciation (at least during the end of the Nantuo glaciation). There are several lithological cycles, including massive diamictite, sandstone/siltstone, mudstone, shales, in Nantuo Formation in South China. These lithological cycles represent the glacial cycles, which include glacially influenced units and non-glacial units (Lang et al., 2018; Shen et al., 2021; Hu et al. 2020). In addition, similar cyclic depositions of Marinoan units have been found in other regions, such as Central Siberia (Chumakov, 2009), Oman (Allen and Etienne, 2008), Norway (Halverson et al., 2004), Southwest China (He et al., 2007), South Australia (Williams et al., 2008), and Scotland (Arnaud and Eyles, 2006). These studies suggest the Marinoan glaciation is dynamic and there might exist open water conditions in the low and middle latitudes.

Here, we added the discussion of the difference between the existing climatic models and the findings in this study. In addition, we presented more geological evidence which support our findings. See Lines 259-272.

(2) The geological background and the interpretation of fossiliferous black shale are not well stated in the manuscript. Although the stratigraphy and sedimentological analyses of the Nantuo Formation have been discussed in previous studies, it is still necessary to display a detailed discussion about the correlation of this black shale layer. I think most readers would be more interested in this story than sedimentological papers. Thus, please do not assume all readers know Nantuo Formation.

Response: Thanks for this good suggestion. Here, we transferred some information to the geological background section of the main text from the supplemental information. See Lines 89-96, 101-104.

(3) It looks like a negative correlation between $\delta^{15}\text{N}$ and $\text{Fe}_{\text{py}}/\text{Fe}_{\text{HR}}$ (Fig. 2b), and there is no pyrite in the lower and upper part of the section. Could pyrite be oxidized?

Response: Thanks for this comment. We note that Fig. 2b does not really show a negative correlation between $\delta^{15}\text{N}$ and $\text{Fe}_{\text{py}}/\text{Fe}_{\text{HR}}$. Nonetheless, we have investigated the pyrite using both a microscope and SEM. However, we did not find any evidence of oxidization of the pyrite. As the Fe concentration is relatively high, we think that sulfate availability was the main factor controlling pyrite content. During deposition of the SBS interval, the sulfate concentration was very low, limiting microbial sulfate reduction. In the middle part of the SBS, the sulfate concentration increased and was sufficient to generate a greater amount of pyrite.

(4) Only a 'blue hole' in mid-latitude looks odd (Fig. 3). I understand that this is the only report showing the presence of open ocean in mid-latitude. I hope authors could make some revision, although I do not have a better solution (sorry!). In fact, I think Figure 4 is not that informative.

Response: Thanks for this comment. Yes, this study is the first to present evidence for open oceans at mid-latitudes. Although it is logical to infer that other open-ocean areas existed at low to mid-latitude areas (see the response above for question #1), we do not know the exact location of these

open-ocean areas. Here, we added two dashed lines in Fig. 3c and noted that other open-ocean areas might have existed in the region bounded by them. In addition, we combined Fig. 3 and Fig. 4 into a single figure (new Fig. 3).

Reviewer #2

Review of Habitable oceanic environment for early eukaryotes extended to mid-latitude at the end of the terminal Cryogenian Snowball Earth

Summary

This study provides nitrogen and carbon concentrations/stable isotope values, in addition to iron speciation, to suggest the presence of low-oxygen bottom water with oxygenated upper water. The studied units also contain preserved fossils, the Songluo Biota, which are reported to be macroscopic eukaryotes. The authors suggest that the presence of

Major points

The discussion of diamictite data should note more clearly that the C and N data most likely reflects a signal of erosion. In addition, while there was some discussion of detrital influence in the bedded unit, there were no geochemical tests of this detrital influence. Measuring Al, for example, can help with this.

Response: Thanks for this important comment and good suggestion. We analyzed the major element concentrations of the samples and added the Al concentrations and CIA values. The new data show that the diamictite consists of about half clays ($\text{Al}_2\text{O}_3 = 7\text{-}8 \text{ wt}\%$) and half carbonate ($\text{CaO} = 10\text{-}15 \text{ wt}\%$). This seems pretty reasonable for glacially generated material as clays and carbonate would have been ground up and mixed together. The CIA values for the diamictite are all ~ 60 (very uniform), which is consistent with well-mixed glacial material. The CIA values for the black shale are 65-70, which is consistent with somewhat greater, yet variable, weathering intensity. See lines 110-111, 115-116.

There are no geochemical measurements to corroborate the interpretation of oxygenated surface water. Enriched N isotope values are certainly consistent with this, but on their own do not robustly demonstrate them.

Response: Thanks for this comment. In fact, most of the geochemical proxies measure bottom water redox, not surface-water redox. One possible proxy that can represent surface water redox is I/Ca, but it only works in carbonates. In this study, our samples are shales so the I/Ca ratio is not applicable. As the atmosphere oxygen concentration rise after the GOE of $\sim 2.4 \text{ Ga}$, the O_2 in the ocean surface increase because the seawater exchanges O_2 with the atmosphere under the open ocean conditions. Furthermore, our $\delta^{15}\text{N}$ data are consistent with oxic surface waters, as noted by the reviewer. In addition, the increased primary productivity (high TOC content) is potential evidence for the oxic surface water. These arguments have been presented in the revised manuscript in greater clarity.

The dataset is a nice addition to the growing global dataset of geochemistry from the Marinoan glaciation. Yet, the discussion is a bit too wide-ranging and expansive. How did the authors decide

where to put the “open water” section in Fig 4c? Could the water belt just be wider at this time? The interpretations are reasonable, but without including previously published data and location/age information it is difficult to say how global the interpretation of oxygenated water is.

Response: Thanks for this comment. Fig 4c (now Fig 3c) is a conceptual model. The general location of the “open water” is based on results from the Songluo section and previously published Cryogenian paleomagnetic data from South China. However, a number of coeval sections globally have yielded similar deposits (i.e., non-glacial sediments with Marinoan glacial units), including in Central Siberia (Chumakov, 2009), Oman (Allen and Etienne, 2008), Norway (Halverson et al., 2004), Southwest China (He et al., 2007), South Australia (Williams et al., 2008), and Scotland (Arnaud and Eyles, 2006). The sedimentological evidence suggests that the Marinoan glaciation was dynamic, and that open-water conditions may have existed widely in low- and mid-latitude regions. As we do not know the exact locations of these potential open-ocean areas, we have added two dashed lines in Fig. 3c that outline the region in which open-ocean conditions might have existed during the Marinoan Ice Age. We have added a brief discussion on this point in lines 259-272.

Methods.

Does WO₃ have any trace C or N in it? Did the authors run samples of just this reagent to test for possible contamination? Same comment for V₂O₅, as previous work (Brauer and Hahne, 2005) have demonstrated that some V₂O₅ has trace N.

Response: Thanks for this comment. In fact, we have analyzed the reagent to test for possible contamination in our lab. There is no detectable C or N signal when we analyzed a blank consisting of 50 mg WO₃ (Macklin, Shanghai, China, PN T818835) using the Elemental Analyzer (Elementar) (Fig. 1) vs the typical C-N signals of the standard samples (Fig. 2). In addition, there is no detectable C and N signals when we analyzed a blank consisting of 10 mg V₂O₅ for nitrogen isotopes (Thermo Scientific, Cambridge, UK, PN 33837510) (Fig. 3) vs the typical signal of 20 μg N (Fig. 4). We infer that the V₂O₅ used by Brauer and Hahne (2005) had a different degree of purity from the V₂O₅ we used in our research. Hence, the WO₃ and V₂O₅ reagents did not influence our C and N concentrations and isotopic compositions. We added this information in the methods section. See Lines 330-331 and 348-349.

Fig. 4. Intensity of 20 µg N under the EA-IRMS (~ 3000 mV).

Minor points

Line 51-54: Why does the presence of eukaryotic fossils imply open water? Are these interpreted as photosynthetic?

Response: As Ye et al. (2015) discussed, these fossils are preserved in thin black shales sandwiched between glacial diamictites deposited in inner-shelf environments of the mid-latitude Yangtze Block. Some of these fossils are interpreted as benthic macroalgae. These fossils represent photosynthetic eukaryotes that lived in the photic zone in open-ocean waters.

Lines 79-83: What is the source of the diamictite? Do the C and N values just reflect provenance?

Response: The glacial diamictite is a mixture of gravel and fine-grained matrix recycled from older rocks. Hence, we believe that the C and N values of the glacial diamictite do not reliably reflect the paleoenvironmental conditions of its deposition, but rather represents a mixture of provenance signal of the material comprising the diamictite, plus some sedimentary C and N formed at the time of deposition.

Line 92: High TOC is not, on its own, a reason to suggest it is resilient to diagenesis. The other metrics, such as lack of correlation between d13C and TOC are much stronger.

Response: Yes, we have presented the correlation between δ¹³C and TOC content. See Lines 127-128 and Supplementary Fig. 1a in the supplementary information. We have added a sentence in the revised manuscript: “Furthermore, there is no significant correlation between δ¹³C_{org} and TOC (Supplementary Fig. 1a), again indicating minimal diagenetic alteration or analytical contamination of δ¹³C_{org}.” (See Lines 127-128)

Line 146: How much can we rely on Fe-speciation in a diamictite? This should probably just reflect the erosional signal from whatever the provenance is. I don’t think you need the end of this

paragraph, as fin

Response: The Fe speciation data in Shen et al., 2022 (Geology) are from the diamictite, sandstone/siltstone, mudstone, carbonate and shale samples. In their supplementary information, these authors noted that “prior to geochemical analyses of Fe-speciation and TOC, the matrix was sampled (finer-grained sediments) in the bulk-rock samples to avoid larger clasts. Then, the matrix was powdered to <74 μm and preserved in a desiccator after being dried at 80°C for 3 h”. In fact, the Fe speciation data of the diamictite probably just represent the erosional signal from whatever the source is. As there are few Fe speciation data for the Nantuo Formation, we think it is desirable to present the Fe speciation data of Shen et al. (2022). Here, we added a note that the Fe speciation data are from the diamictite matrix, sandstone, mudstone and shales. See Lines 175-177.

Line 176: Add “net” before “isotopic fractionation”, as there is an isotope effect during nitrification.

Response: Thanks. Added.

Line 185: In this paragraph, comparing the current work to previous work, it is important to note the units in the Ghaub and Mineral Fork formation are not diamictite, but rather finely bedded units interbedded within a massive diamictite.

Response: Thanks for this suggestion. For the Ghaub Formation, the detailed section at Fransfontein highlights samples taken from only finely laminated beds without ice-rafted debris or turbidites. Samples at Bethanis were taken at about 10 m spacing throughout the section (Johnson et al., 2017). For the Mineral Fork Formation, most samples are from the finely bedded units interbedded with massive diamictite, and two samples are massive till with cm-sized clasts and coarse sandstone (Johnson et al., 2022). Hence, we add a note “Here, most of these studied samples are from the finely bedded units interbedded with massive diamictite in the Ghaub and Mineral Fork Formation” in Lines 233-235.

Methods.

Does WO_3 have any trace C or N in it? Did the authors run samples of just this reagent to test for possible contamination? Same comment for V_2O_5 , as previous work (Brauer and Hahne, 2005) have demonstrated that some V_2O_5 has trace N.

Response: See above for response.

Figure 2. What are the uncertainties on Fe speciation data?

Response: The Fe speciation value is calculated as the ratio of several different Fe species (total (Fe_T), iron oxides (Fe_{ox}), magnetite (Fe_{mag}), carbonate-associated iron (Fe_{carb}), and pyrite iron (Fe_{py}). In the analytical process, the test of each Fe species has a certain error, which is described in the methods (Lines 360-362). The error for $\text{Fe}_{\text{HR}}/\text{Fe}_T$ and $\text{Fe}_{\text{py}}/\text{Fe}_{\text{HR}}$ can be calculated from the analytical errors of Fe_T , Fe_{ox} , Fe_{mag} , Fe_{carb} , and Fe_{py} , using error propagation equations.

Figure 3. Do you think it's really like an oxygen minimum zone? Would a more continuous redoxcline also be a possibility?

Response: Thanks for this comment. Yes, we agree that a continuous redoxcline is possible and perhaps even more likely than an OMZ, and we have modified Figure 3c accordingly. In fact, whether an OMZ existed in the Cryogenian remains an open question. In the open ocean, the downward flux of organic matter from the surface layer generally results in a maximum intensity of remineralization at upper bathyal water depths (i.e., 200-1000 m, corresponding to modern OMZs), but with low atmospheric O₂ levels and inadequate ventilation of the deep ocean, it is possible that a modern-like OMZ did not exist during the Cryogenian.

Figure 4. Can you somehow combine figure 3 and 4? Could the right-hand columns basically replicate what's already in figure 3? This could be a nice summary, instead of two separate figures.

Response: Thanks for this suggestion. In the revision, we combined Fig. 3 and Fig. 4 into a single figure (new Fig. 3).

References cited above

- Allen, P. A. & Etienne, J. L. Sedimentary challenge to Snowball Earth. *Nat. Geosci.* 1, 817-825 (2008).
- Arnaud, E. & Eyles, C. H. Neoproterozoic environmental change recorded in the Port Askaig Formation, Scotland: Climatic vs tectonic controls. *Sediment. Geol.* 183, 99-124 (2006).
- Bräuer, K. & Hahne, K. Methodical aspects of the ¹⁵N-analysis of Precambrian and Palaeozoic sediments rich in organic matter. *Chem. Geol.* 218, 361-368 (2005).
- Chumakov, N. M. Neoproterozoic glacial events in Eurasia. In: Gaucher, C., Sial, A. N., Frimmel, H. E. & Halverson, G. P. (Eds.), Neoproterozoic-Cambrian Tectonics, Global Change and Evolution: A Focus on South Western Gondwana, *Developments in Precambrian Geology* 16, pp. 389-403 (2009).
- Halverson, G. P., Maloof, A. C. & Hoffman, P. F. The Marinoan glaciation (Neoproterozoic) in northeast Svalbard. *Basin Res.* 16, 297-324 (2004).
- He, J. et al. Neoproterozoic sequence stratigraphy and correlation in Quluqtagh area, Xinjiang. *Acta Petrol Sin.* 23, 1645-1654 (2007) (in Chinese with English Abstract).
- Hyde, W. T., Crowley, T. J., Baum, S. K. & Peltier, W. R. Neoproterozoic 'snowball Earth' simulations with a coupled climate/ice-sheet model. *Nature* 405, 425-429 (2000).
- Johnson, B. W., Poulton, S. W. & Goldblatt, C. Marine oxygen production and open water supported an active nitrogen cycle during the Marinoan Snowball Earth. *Nat. Commun.* 8, 1316 (2017).
- Johnson, B. W., Mettam, C. & Poulton, S. W. Combining nitrogen isotopes and redox proxies strengthens paleoenvironmental interpretations: examples from Neoproterozoic Snowball Earth sediments. *Front. Earth. Sci.* 10, 745830 (2022).
- Lang, X. et al. Cyclic cold climate during the Nantuo Glaciation: evidence from the Cryogenian Nantuo Formation in the Yangtze Block, South China. *Precambrian Res.* 310, 243-255 (2018).

- Micheels, A. & Montenari, M. A snowball Earth versus a slushball Earth: Results from Neoproterozoic climate modeling sensitivity experiments. *Geosphere* 4, 401-410 (2008).
- Shen, W. et al. Secular variation in seawater redox state during the Marinoan Snowball Earth event and implications for eukaryotic evolution. *Geology* 50, 1239-1244 (2022).
- Williams, G. E., Gostin, V. A., McKirdy, D. M. & Preiss, W. V. The Elatina glaciation, late Cryogenian (Marinoan epoch), South Australia: sedimentary facies and palaeoenvironments. *Precambrian Res.* 163, 307-331 (2008).
- Ye, Q. et al. The survival of benthic macroscopic phototrophs on a Neoproterozoic snowball Earth. *Geology* 43, 507-510 (2015).

Reviewer #1 (Remarks to the Author):

I am satisfied with the revised manuscript. It seems authors have well addressed my comments and suggestions of the original submission. Thus, I think the manuscript could be accepted for publication.

Reviewer #2 (Remarks to the Author):

Thank you for taking the time to provide thoughtful responses and additional geochemical measurements! The addition of tests to make sure the reagents are not contributing to the N signal are great, and the AI data is helpful as well.

Please cite Ader et al., 2016 in the figure caption of Fig 3, since these reconstructions are based on figures from that publication.

Response to the referees (NCOMMS-22-40658A)

Reviewer #1 (Remarks to the Author):

I am satisfied with the revised manuscript. It seems authors have well addressed my comments and suggestions of the original submission. Thus, I think the manuscript could be accepted for publication.

Response: Thanks for your positive comments.

Reviewer #2 (Remarks to the Author):

Thank you for taking the time to provide thoughtful responses and additional geochemical measurements! The addition of tests to make sure the reagents are not contributing to the N signal are great, and the Al data is helpful as well.

Please cite Ader et al., 2016 in the figure caption of Fig 3, since these reconstructions are based on figures from that publication.

Response: Thanks for your positive comments. We cited Ader et al., 2016 in the figure caption of Fig 3.